# Is COVID-19 as Lethal as the Spanish Flu? The Australian Experience in 1919 and 2020 and the Role of Nonpharmaceutical Interventions (NPIs)

**DOI:** 10.3390/ijerph21030261

**Published:** 2024-02-23

**Authors:** Marika Vicziany, Leon Piterman, Naiyana Wattanapenpaiboon

**Affiliations:** 1Monash Asia Initiative, Faculty of Arts, Monash University, Caulfield, VIC 3145, Australia; tikky.wattanapenpaiboon@monash.edu; 2Department of General Practice, School of Public Health and Preventive Medicine, Monash University, Clayton, VIC 3800, Australia; leon.piterman@monash.edu

**Keywords:** COVID-19, Spanish Flu, lethality, hospital admissions, death rates, nonpharmaceutical interventions (NPIs), pandemic response, quarantine, lockdowns

## Abstract

We conducted a comparative historical study to interrogate Professor Peter Doherty’s warning to Australians in April 2020 that ‘COVID-19 is just as lethal as the Spanish flu’. We identified the epicentres of both pandemics, namely, metropolitan Sydney in 1919 and metropolitan Melbourne in 2020 and compared the lethality of the Spanish Flu and COVID-19 in these two cities. Lethality was measured by the number and rate of hospital admissions, death rates, age-specific death rates and age-standardised mortality rates (ASMRs). Using these measures, we demonstrated the strikingly different waves of infection, their severity at various points in time and the cumulative impact of the viruses by the end of our study period, i.e., 30 September in 1919 and 2020. Hospital admissions and deaths from the Spanish Flu in 1919 were more than 30 times higher than those for COVID-19 in 2020. The ASMR per 100,000 population for the Spanish Flu was 383 compared to 7 for COVID-19: The former was about 55 times higher than the latter. These results suggest that the Spanish Flu was more lethal than COVID-19. Professor Doherty’s warning was perhaps taken seriously and that partly explains the findings of this study. Containing infection in 1919 and 2020 threw the burden on nonpharmaceutical interventions (NPIs) such as ‘protective sequestration’ (quarantine), contact tracing, lockdowns and masks. It is likely that the persistent and detailed contact tracing scheme provides the best possible explanation for why NPIs in 2020 were more effective than in 1919 and therefore contributed to the lower lethality of the COVID-19 pandemic in its first year.

## 1. Introduction

Professor Peter Doherty, a Nobel-prize laureate and world authority on viruses, warned Australians in April 2020 that ‘COVID-19 is just as lethal as the Spanish flu’ [1]. His concern was based on COVID-19 having a basic reproduction ratio (R_0_) of 2–3 reported at that time. However, the R_0_ for Spanish Flu was not commented on. Given this, we decided to undertake a comparative history of the lethality of COVID-19 in 2020 and the Spanish Flu in 1919 using Australia as a case study. What did the waves of infection look like, how quickly did the pandemics spread, and most important of all, does the evidence for 1919 and 2020 show that both pandemics were equally lethal as Professor Doherty suggested? The current ‘pandemic era’ which began around 1900 has taught us to anticipate ‘inconsistent behaviour’ in the number, nature and duration of waves of infection and mortality [2,3]. Strict maritime quarantine protected Australia from Europe’s terrible pandemic waves in late 1918, with the result that Australia had only two waves [4]. This comparative study employs original data for the Spanish Flu and COVID-19 from the first confirmed cases of infection in January 1919 and 2020. Our study period runs from 25 January to 30 September in both years. One important caveat to be added is that without laboratory tests for the virus and modern contact tracing in 1919, there is no certainty about the exact dates of first cases of Spanish Flu in Australia or Sydney.

The comparison between 1919 and 2020 can be justified not only because of the benefits to be gained from historical knowledge and the need for future pandemic planning, but also because there were no pharmaceuticals in either 1919 or 2020. The only possible responses rested with nonpharmaceutical interventions (NPIs) in the general community. While we recognise the essential roles of NPIs in the case of healthcare workers in hospitals, other clinical settings and aged care/disability facilities, their contributions to preventing transmission of the virus in these settings are discussed elsewhere [5].

It is salutary to remind ourselves that inequality in developed countries today, including Australia, has reached the old levels of the pre-Keynesian revolution of the 1940s [6]. This is reflected in the studies on the COVID-19 pandemic showing a correlation between a higher incidence of COVID-19 infection in metropolitan Melbourne in 2020 and ‘larger proportions of people who were unemployed, without paid leave benefits, or experiencing mortgage or rent stress, and in areas with higher population and housing density or larger proportions of people who speak languages other than English at home’ [7]. The report of the Parliamentary Inquiry of 2021 confirmed this: in metropolitan Melbourne, ‘The five councils that had the most active COVID-19 cases as at 1 December 2020 are amongst the 10 most disadvantaged councils in Victoria’ [8].

Despite the enormous pressures on Melbourne’s health and contact tracing systems, in 2020, we learnt how to slow down or ‘flatten-the-curve’ and reduce the lethality of COVID-19 using NPIs, while we waited for medical science to catch up with our needs by developing new drugs and vaccines. Paradoxically, the arrival of antivirals and vaccines from 2021 onwards has coincided with a great increase in deaths directly or indirectly caused by or related to COVID-19. The increasing abandonment of NPIs has, we argue, contributed to this.

## 2. Materials and Methods

The following four steps were taken for our data analysis.


*Step 1: Identifying the epicentres of the two pandemics*


Identifying the epicentres of both pandemics, namely, metropolitan Sydney in 1919 and metropolitan Melbourne in 2020, was based on two measures of lethality—the number and rate of hospital admissions and deaths for the respective calendar years. We then evaluated the waves of infection and compared their severity at various points in time and the cumulative impact of the viruses. It has been estimated that the total number of deaths caused by the Spanish Flu in Australia were up to 15,000 in the single year of 1919 [9]. However, for this study, we have cited the more conservative figure of 11,552 given in the *Commonwealth Official Year Book 1901–1919* [10]. For the capital cities. we used the local state reports for the relevant calendar years [11,12] because it was the states that reported their estimates to the Commonwealth Government. We also used the Australian Government Department of Health report on COVID-19 [13] for the comparisons (Table 1).

Due to the lack of publicly accessible information required for this step, we have made assumptions to justify the legitimacy of proxy numbers presented in Table 1. Even though we sought to generate comparisons for the whole calendar years 1919 and 2020, we were compelled to use the figure of 3902 deaths for Sydney in 1919 which covered the period up to 30 September not 31 December 1919. Using this proxy number, however, is reasonable because (a) by the end of September 1919, deaths from Spanish Flu in Sydney were down to single figures and did not rise again; (b) the deaths reported for Melbourne in 1919 were for the whole calendar year but the pandemic was officially declared to have ended in that city on the 8 September 1919 [12], suggesting that as with Sydney, the pandemic had reached single figures. In other words, after 30 September 1919, it is reasonable to assume that the differential impact of the Spanish Flu on the two cities did not change beyond what is stated in Table 1.

Information on the total number of COVID-19 deaths in Australian capital cities at the end of December 2020 was not publicly available. However, it has been reported that deaths in the major cities of Australia accounted for about 87% of total deaths [15]. We have, therefore, used the total number of COVID-19 deaths for each of the states as proxies—i.e., 54 deaths in NSW and 820 in Victoria (Table 1). The relatively very small number of COVID-19 deaths in NSW indicates that Victoria carried the burden of COVID-19 lethality in 2020 (820 out of the national total of 909). Given that 76.5% of the population in Victoria resides in Melbourne (Table 1), it is reasonable to assume that Melbourne was Victoria’s epicentre for the pandemic.


*Step 2: The prevalence of infection in the general community*


The second step in our comparison of lethality in Sydney and Melbourne during the two pandemics focuses on the incidence or prevalence of infection in the general community. Detailed data about the weekly number of COVID-19 cases in the community in 2020 were requested from Victoria’s Department of Health and Human Services (hereafter DHHS); however, we were unable to use them in our comparisons because there were no matching data for the 1919 Spanish Flu. Attempts to keep track of daily infections in 1919 were abandoned in the early stages of the pandemic [16]. Moreover, we have no scientific way of knowing the actual prevalence of infection because the human influenza virus was not identified until 1933 and the genome of the H1N1 virus was only reconstructed in 1997–2005 [17]. However, Armstrong, the NSW Director General of Public Health at the time, conducted an investigation and concluded in his 1920 report that altogether, about 290,000 or 36.6% of Sydney residents had caught the Spanish Flu in 1919 [18]. With the circumstances facing the Sydney health authorities in 1919, the estimate of the extent of infection in the metropolitan area appears to be reasonable, yet we have also questioned it.


*Step 3: Hospital admissions as a proxy for the analysis of the waves of infection*


While no data exist for infections of Spanish Flu in the general community in 1919, we have used the weekly data about hospital admissions given by Armstrong (the Director General of Public Health) in his *Report on the Influenza Epidemic in New South Wales in 1919* [18] as a proxy for the waves of community infection. Armstrong’s report ran from 25 January or the week ending 1 February to 30 September, a total of 35.5 weeks. The start of the time series reflected the dates for the first known Spanish Flu infections in Sydney. Fortuitously, the 25 January was also the date of the first confirmed infection by COVID-19 in Melbourne in 2020. Information about hospital admissions in metropolitan Melbourne was provided by the DHHS in response to our request for a detailed breakdown of data by each week of the study period and by all of the ‘Local Government Areas’ for metropolitan Melbourne [19].

Using these sources about hospital admissions as proxies for trends in community infections during the pandemics of 1919 and 2020 enables us to assess the fluctuations in weekly confirmed cases and the cumulative cases at the end of our study period, 30 September 1919 and 2020. However, the lack of reliable information about the cases of infection in the Sydney community makes it impossible to calculate the true case fatality rates (CFRs) in 1919.


*Step 4: Analysis of the trends in weekly death rates*


The Director General of Public Health’s *Report on the Influenza Epidemic in New South Wales in 1919* also gave us information about the weekly deaths in Sydney in 1919 [11], while the DHHS provided us with data about the weekly COVID-19 deaths in Melbourne in 2020 [19]. However, some minor adjustment to these data sets was needed. While the influenza report [11,18] was comprehensive and detailed for our study period (25 January to 30 September 1919 and 2020), the reporting dates of hospital admissions and deaths were slightly different: weekly hospital admission reports began on Saturday 1 February, but weekly death rates began on Tuesday 4 February. Given that we were not investigating the time lag between admissions and deaths, we adjusted the reporting date for deaths in line with the reporting date for hospital admissions, i.e., the first reporting date for deaths in our comparison is now given as Saturday 1 February instead of Tuesday 4 February. These adjusted dates facilitated the comparisons between the weekly hospital and death rates in Sydney in 1919 and Melbourne in 2020.

Given the unique nature of the Sydney data and the absence of alternative or more disaggregated sources, we took these adjusted data sets as our starting point and matched the Melbourne data with them. The matching time series created for COVID-19 in 2020 was based on disaggregated data on Local Government Areas for both hospital admissions and crude death rates from Victoria’s DHHS [19]. Data for the week ending 1 February mean that our study period captures trends in the preceding week for both hospital admissions and crude deaths in Sydney and Melbourne. In both 1919 and 2020, the 1 February was a Saturday, a coincidence that ensures that our weekly timeframes are comparable. In summary, the comparisons between the pandemic in Sydney and Melbourne both cover the study period of some 35.5 weeks from 25 January to 30 September.

Understanding mortality patterns is a complex matter, and during the COVID-19 pandemic in Melbourne, a great public debate emerged about who needed to be vaccinated first. Apart from seriously immuno-compromised people, the priority given to specific age groups became important. Given this, we have also calculated age-specific death rates for the Spanish Flu in 1919 and COVID-19 in 2020 using information from various sources including the 1921 and 2021 Australian Censuses (Australian Bureau of Statistics, ABS) official reports [14,20,21]. A data set on COVID-19 deaths by sex and age was constructed from surveillance data that the DHHS in Victoria made available for public download [22]. As previously mentioned, the information on COVID-19 deaths was available for the whole state of Victoria, not metropolitan Melbourne, leaving us with no option but to use the Victorian data as proxy. The numbers of deaths caused by the Spanish Flu in 1919 were originally reported in 5-year age groups [20], which we collapsed into 10-year age groups. This was to allow for a simpler comparison of age-specific death rates between the two pandemics. We also calculated the age-standardised mortality rate (ASMR) for the Spanish Flu in Sydney and COVID-19 in Melbourne with adjustment to the WHO standard population age structure [23].

## 3. Results

### 3.1. Crude Death Rates in the Epicentres of Sydney (1919) and Melbourne (2020)

Using the percentage of national deaths, Table 1 shows metropolitan Sydney as the epicentre of the Spanish Flu in 1919 and metropolitan Melbourne of COVID-19 in 2020: 3902 deaths representing 33.8% of Australian deaths caused by the Spanish Flu in 1919 occurred in Sydney and 820 deaths or 90.2% of Australian deaths caused by COVID-19 were recorded in Melbourne in 2020. The table also yields estimates of the crude death rates per 100,000 population in these two cities: 471 for Sydney in 1919 and 16.1 for Melbourne in 2020. Another way to express this is to say that mortality from the Spanish Flu in Sydney in 1919 caused almost 30 times more deaths than COVID-19 in Melbourne in 2020.

### 3.2. Contrasting Patterns of Hospital Admissions as a Proxy for Infections

It has been estimated that the cumulative infections in Sydney, with a population of 828,700 in 1919, were about 290,000 [18] or 34,995 cases per 100,000 population. This compares with 20,368 cumulative cases of COVID-19, using the state of Victoria’s figure as a proxy [13], or 401 cases per 100,000 population for metropolitan Melbourne with a population of about 5.1 million. The comparison suggests that the infection rate in the general community in Sydney in 1919 was about 87 times greater than that in Melbourne in 2020.

The NSW Department of Health reported that by 30 September 1919, some 14,164 people had been hospitalised in Sydney compared with 2518 hospital admissions in Melbourne (Figure 1A). Cumulative Melbourne hospital admissions were only 18% of those in Sydney in 1919. The great stress on Sydney’s hospitals is revealed in Figure 1A,B, showing that hospital admissions in Sydney exceeded those in Melbourne 5.6 times. The cumulative hospital admission rates show an even greater disparity between Sydney and Melbourne in 1919 and 2020: 1709 relative to 50 per 100,000 population (Figure 2). Sydney admissions were 34 times higher than those in Melbourne.

Figure 1A,B show that the Sydney pandemic was characterised by two very high and distinctive peaks, while Melbourne’s crisis took the form of one major wave from mid-July to mid-September, with only one peak. The weekly hospital admissions in Sydney first peaked in the week ending 11 April, with 1025 admissions, and then again in the week ending 20 June with 1315 admissions, while in Melbourne, they peaked in the week ending 8 August, with 419 admissions. Given the much larger population of Melbourne in 2020 relative to Sydney in 1919, Figure 1B shows that the dramatic differences between the two pandemics was even greater than the crude numbers suggest: per 100,000 population, the highest weekly hospital admission rates in Sydney were 124 and 159 per 100,000 population for each peak compared with Melbourne’s peak on 8 August of about 8.

Figure 2 shows the cumulative hospital admission rates (per 100,000 population) at any point in time for Sydney in 1919 and Melbourne in 2020—that is, the number of new admissions in week X plus all admissions in the weeks prior to that point. Cumulative weekly admission rates in Sydney in 1919 started at a higher point than in Melbourne in 2020 and continued to be higher for the rest of the study period. At the end of September, the cumulative hospital admission rates in Sydney were 1709 compared with Melbourne’s of 50.

### 3.3. Contrasting Patterns of Death

In the analysis of death patterns, the phases of each pandemic were defined by the number of deaths per week that were reported as zero, single- (1–9), double- (10–99) or triple-digit (100+) numbers. The terminology of single-, double- and triple-digits was often used in government briefings and media reports in 2020 to inform the general public about how serious the situation was.

The results in Table 2 confirm the high lethality of Spanish Flu in Sydney and capture some of the desperation that Sydney experienced in 1919. The first death had been reported on 4 February 1919 and weekly deaths in single-digit numbers were reported for the following 7 weeks. After a week of double-digit weekly deaths (Phase 2) mortality surged to weekly triple-digits for 6 weeks (Phase 3) followed by a decline into double-digit figures for 5 weeks (Phase 4). However, instead of continuing to fall to single-digits, mortality surged again to triple-digits for another 5 weeks (Phase 5) before dropping to double-digit numbers and eventually to single numbers by 30 September 1919. The total of 11 weeks of triple-digit (100+) deaths in Phases 3 and 5 represented 31% of the pandemic’s duration in Sydney.

By contrast, Melbourne experienced far less trauma because the period with triple-digit weekly deaths lasted only 3 weeks (Phase 4). Even more markedly, despite recording the first confirmed positive case of COVID-19 on 25 January, Melbourne’s rising case numbers (infections) did not generate any deaths for the first 15 weeks or 42% of the duration of the COVID-19 pandemic. Taking all of the weeks in Phases 1 and 2 together, when weekly deaths were either zero or single-digit numbers, we can say that these 24 weeks up to 11 July demonstrated the low impact of the virus by causing only 32 deaths.

Cumulative death rates shown in Figure 3 depict the relatively devastating situation in Sydney in 1919. There was no death reported in the first week of the Spanish Flu pandemic; the first death was recorded in Sydney for the week ending 4 February 1919. Then, there was a surge involving 767 deaths during the following 11 weeks. This represents a jump in the cumulative death rates from 0.1 to 92.5 per 100,000 population. After that, within 8 weeks, the cumulative weekly death rate doubled to 201 and doubled again in the following 4 weeks to 420. Then slowly, the cumulative death rate reached 471 per 100,000 population on 30 September 1919. A different pattern can be seen for COVID-19 in 2020, where the first confirmed case in Melbourne was recorded on the 25 January 2020, but zero or only single-digit weekly deaths were recorded during the first 17 weeks of the pandemic. As shown in Figure 3, weekly deaths in Melbourne began to rise during the week ending 21 March 2020, but a plateau was quickly reached and maintained for some 17 weeks until mid-July, with total cumulative deaths coming to only 32 on 11 July. In other words, during the first six months of the pandemic, total deaths in Melbourne from COVID-19 were exceptionally low. After that, there was a considerable surge; within three weeks from 11 July, the cumulative death rates rose from less than 1 to 15.5 deaths per 100,000 population. Another plateau was then reached, and this continued till the end of September 2020.

As illustrated in Table 2 and Figure 3, the speed with which the Spanish Flu pandemic unfolded in Sydney contributed to the high cumulative death rate of 471 per 100,000 population by the end of September 1919. This rate was about 30 times higher than that of Melbourne (15.5 per 100,000 population) for the same period from the week ending 1 February to 30 September in 2020.

The contrasting pattern of deaths between the Spanish Flu and COVID-19 experienced in Australia can also be demonstrated using the measure of death rates in certain age groups, as shown in Figure 4. The elevated mortality caused by COVID-19 in 2020 is a story mainly about the experiences of the over 60-year-old age groups, with dramatic increases for those 80 years and older. A very different scenario can be seen for the Spanish Flu, with a disproportionate impact of mortality on the cohort aged 20–49 years (Figure 4), a group that constituted the bulk of Sydney’s working population of men and women. This is one of the best-known aspects of the Spanish Flu pandemic of 1919 [16].

The age-specific death rates for Sydney and Melbourne allowed us to calculate the ASMRs for these two cities in 1919 and 2020. The age-specific death rates were adjusted to the WHO standard population age structure [23]. The result showed that the ASMR for the Spanish Flu was 383 deaths per 100,000 population and for COVID-19, it was 7 per 100,000 population. By this measure, the lethality of the Spanish Flu in Sydney was about 55 times greater than COVID-19 in Melbourne.

### 3.4. Implementation of NPIs and Their Impact on the COVID-19 Pandemic

Figure 5 shows the total number of weekly COVID-19 cases in Melbourne relative to hospital admissions, ICU admissions and deaths during our study period. It depicts the two waves in the first year of the COVID-19 pandemic experienced by Melbourne residents. The first, relatively small wave was driven by international arrivals (foreign visitors and returning Australian citizens), while the second, much larger wave began when international arrivals passed on their infections to the personnel in charge of the quarantine hotels. From there, the infection spread through the workforce and ultimately found its weakest targets in the aged care homes of metropolitan Melbourne.

The rise and fall in cases in Melbourne drove the key policy decisions about interventions by the State Government of Victoria and local governments. These policies took the form of key NPIs, namely, mandatory restrictions of increasing severity (Stages 1–4), two major lockdowns lasting a total of 154 days, testing blitzes and mandatory face coverings. Figure 5 includes the timeline for these NPIs—they drove down the level of infections in the community or what the public health officials called ‘flattening-the-curve’, as observed in this and other studies [24].

In summary, the above results show that the Spanish Flu in Sydney in 1919 was between 30 and 55 times more lethal than COVID-19 in Melbourne in 2020.

## 4. Discussion

The evidence in this study indicates that Professor Doherty’s statement that ‘COVID-19 is just as lethal as the Spanish flu’ [1] does not appear to hold for any measures of lethality employed in our study. Using original sources and publicly accessible data, we have demonstrated that the cumulative hospital admission rates, death rates and ASMRs from the Spanish Flu in Sydney in 1919 were markedly higher than those from COVID-19 in 2020. The greater lethality of the Spanish Flu was generated by elevated mortality rates that were more persistent and lasted longer than was true of COVID-19 in Melbourne in 2020. Therefore, instead of being ‘just as lethal as the Spanish Flu’, our findings suggest that the COVID-19 pandemic in Melbourne in 2020 was less lethal. Had Melbourne hospitals been confronted by the same number of hospital admissions as Sydney in 1919, the health system would have been overwhelmed.

The dramatic difference between the Spanish Flu and COVID-19 scenarios is not unique to Australia; Spanish Flu cases in Switzerland were 40 times higher than confirmed COVID-19 cases in 2020 [25]. These estimates are counterintuitive given that relative to the Spanish Flu, COVID-19 appears to have a greater propensity to mutate, resulting in greater infectiveness and a longer lifespan because of the ongoing cycle of mutations [26,27]. Of course, without any genomic data for 1918–1919, we can only speculate about the nature of the influenza virus and its various strains at that time. In 2020, the high infectiveness of COVID-19 was identified as being linked to its longer incubation period of between 4 and 12 days [28]. The incubation period of the Spanish Flu virus was very much shorter, from a few hours up to one or two days [25,29], but there is insufficient explanation of the true characteristics of the virus that caused the 1918–1919 pandemic.

The severity of the Spanish Flu has never been fully explained despite the research by Tumpey and colleagues on the virulence of the reconstructed Spanish Flu H1N1 virus in 2005 [30]. Moreover, the influenza virus may not have been the primary driver of mortality; perhaps it was secondary infections from a range of ‘bacterial pneumopathogens’. These worked in tandem with the H1N1 virus amongst a population that had no access to antibiotics which did not yet exist [2]. As a result of the deprivation experienced by all sections of society during the First World War, the immunity of Australians was also very low, making them susceptible to viral and bacterial pathogens.

The age-specific nature of mortality from the Spanish Flu may have been another factor driving the severity of the 1919 pandemic relative to the COVID-19 pandemic in 2020. As shown in Figure 4, the most severely affected age group in the Sydney population was the 20- to 49-year-old cohort. According to the 1921 Census, this group represented about 47% of the total population and perhaps the most economically productive members of the workforce. By contrast, the elevated mortality rates from COVID-19 were experienced mainly by the over 80-year-old cohort in Melbourne, whose proportion was about 3.8% of the Victorian population. It is evident that the Spanish Flu in 1919 was more economically devastating than COVID-19 in 2020 partly because the flu virus preferred younger people who represented a larger proportion of the country’s population at the time compared to the smaller proportion of elderly people in the general population in Melbourne in 2020.

The lesser lethality of COVID-19 in 2020 is not, however, an index of lesser human suffering. During the first year of the COVID-19 pandemic, Melbourne’s victims of infection largely dealt with their own sickness in home isolation. At the end of our study period, 30 September 2020, this ‘hidden health crisis’ meant that of the 305 active cases in Victoria, only 44 (about 15%) were in hospital, including 6 in intensive care [31]. The remaining 85% of active COVID-19 cases were nursing themselves at home, many of them using the option of telehealth with health professionals. Thousands more of symptomatic and asymptomatic Australians also suffered the physical and psychological effects of COVID-19 alone. Throughout the year, health professionals were being called upon to become directly involved in home-based or clinic-based primary healthcare of COVID-19 patients [32], despite official policies that placed the management of the pandemic in the hands of hospitals and clinics.

The difference between home nursing in 1919 and 2020 is that Melbourne’s ‘hidden health crisis’ in 2020 did not distort the data we have on COVID-19 deaths or hospital admissions. This is because mortality from COVID-19 was concentrated in aged care homes where many of the oldest and most vulnerable Australians lived. On 19 September 2020, about 74% of deaths from SARS-CoV-2 had occurred amongst those ‘living in aged care homes at the time of their deaths, although many died in hospital’ [33]. Cases of infection in the ten worst-affected aged care homes in Victoria on 31 July 2020 came to 49% of residents and 38% of staff [34,35]. The concentration of deaths amongst older persons and within particular facilities means that the data on hospital admissions and deaths are relatively complete and accurate.

Much can be learned about how changing human behaviour can dampen the impact of pandemics [36]. In 1919, the many measures that were taken to reduce the spread and severity of the Spanish Flu included the closure of international borders, the lockdown of schools and other public facilities, the wearing of face masks and health measures such as public inhalations, vaccination and the distribution of food aid. These NPIs were useful but insufficient to reduce the high levels of mortality and morbidity that were experienced [37,38]. Despite the imperfections of NPIs, we should not dismiss the importance of early and timely nonpharmaceutical interventions because without any changes in human behaviour, the outcomes of the Spanish Flu in Sydney might have been much worse.

### 4.1. Protective Sequestration as an NPI

‘Protective sequestration’ or quarantine was highly successful in keeping the Spanish Flu virus at bay in Australia. As an isolated island continent, stringent port regulations made that possible, with the result that the most savage influenza strains of 1918 did not enter the country and Australia escaped the first global waves of the pandemic. By late January 1919, the situation changed with the outbreak of the Spanish Flu in Melbourne and Sydney, the detailed circumstances of which have not yet been fully explained. Once the pandemic burnt itself out, memories of the Spanish Flu dimmed quickly despite the huge number of deaths. The debates about how to control the pandemic, the protests against masks and the policy changes that defined the responses of other countries to influenza became features of Australia’s encounter with H1N1 in 1919. There has been no study of the Australian experience comparable to the American research by Markel and others [39,40], but there is no reason to think that quarantine in Australia during the Spanish Flu did not make an equally important contribution to dampening the waves of infection.

In 2020, Australia’s quarantine system needed to respond to a much more complex international economy, with frequent arrivals from many global destinations at many entry points. On 1 February 2020, the first ban was introduced preventing all foreign nationals who had left mainland China from landing in Australia until 14 days had passed since they left the People’s Republic of China [41]. It was widely assumed that COVID-19 originated in Wuhan, mainland China, as a result of either natural or unnatural causes [42]. On the 20 March 2020, Australia closed its international borders to all but Australian citizens, permanent residents and returning family members [41]. By then, Stage 1 restrictions had been introduced four days earlier, Stage 2 restrictions followed on 25 March, and Stage 3 restrictions on 30 March (Figure 5). Restrictions were not eased till 11 May and then re-introduced on 30 June. These rapid responses to rising cases of infection began with bans against social activities in order to curtail the transmission of COVID-19 in public places. For example, on 22 March, the Victorian government ordered the closure of pubs and bars, entertainment venues, religious gatherings, and cafes and restaurants in Melbourne. A few exceptions were allowed such as the purchase of takeaway food and drink, and very small groups could meet at funerals and some religious events where the ‘1 person per 4 square meters’ of social distancing applied [41].

By the end of March 2020, Australia became one of 12 countries to introduce quarantine hotels as a way of detaining all international arrivals for 14 days of surveillance and testing for COVID-19. However, the system was flawed, with 20 known breaches of quarantine and complaints by international arrivals of the poor condition of many of the hotels and how they felt at risk of catching COVID-19 and other infections [43]. In May and June, breaches at the Rydges and Stamford Hotels in Melbourne led to the second wave of COVID-19 infection spreading through the community, especially into the aged care sector. The former hotel was linked to about 90% of COVID-19 fatalities in Melbourne in 2020 and the latter to the remaining 10% [44]. Our certainty about these transmission routes is based on the genomic sequencing scheme that was developed to identify clusters of different variants of the virus in the community. The effectiveness of Australia’s quarantine system in 2020 has yet to be systematically assessed but it is reasonable to suggest that despite the inadequacies, the system was a great advance on what was possible in 1919 because testing for the SARS-CoV-2 virus was possible.

The second wave of the COVID-19 pandemic brought about the introduction of mandatory mask wearing on 22 July and Stage 4 restrictions on 2 August (Figure 5). The heightened restrictions came a month after the authorities had already imposed the second lockdown of metropolitan Melbourne from 9 July. This lockdown lasted until 27 October, totalling 111 days, making Melbourne the most-locked down city in the world [45]. The July 2020 lockdown was one of six implemented during the first two years of the pandemic [46]. On 2 August 2020, the Victorian government declared a state of disaster, imposed Stage 4 restrictions and a curfew from 8 p.m. to 5 a.m. Six weeks later, on 14 September, the curfew was eased to last from 9 p.m. to 5 a.m. and then removed on 28 September [47].

The lockdowns included the specific isolation and temporary closures of particular facilities where infectious clusters had been identified; for example, on 1 August 2020, there were 823 active cases in aged care homes, 377 in public housing towers, 286 amongst meatworkers, 184 in the Al-Taqwa College and several cases in retail outlets [48]. Through these means, often enforced by police and members of the Australian Defence Force, the Victorian government sought to contain the spread of infection at a time when little was known about the virus and when pharmaceutical solutions were not available. Government intervention did not waiver despite many justifiable complaints about the imperfect implementation of these policies. The benefits of imposing ‘protective sequestration’ in Melbourne through targeted lockdowns were partially compromised by the feeling amongst some communities that they were under attack. The reason for more rather than less intervention was that without vaccines and antivirals, ‘an abundance of caution’ was necessary. Health professionals in Melbourne claimed that these NPIs could help to minimise the risks of cross-infection, particularly in facilities with high population densities where physical distancing was impossible This rationale was logical, but the lockdowns could have been implemented in better ways and with communications and ‘outreach to multicultural communities’ who did not speak English [49].

### 4.2. Contact Tracing and Genomic Sequencing as Part of the Protective Sequestration Scheme

In 2020, ‘protective sequestration’ was implemented on a scientific basis thanks to our capacity to test for the presence of SARS-CoV-2. This led to contact tracing and the genomic sequencing of the SARS-CoV-2 virus. This in turn enabled governments in Australia to generate detailed maps of the outbreaks of infection and the geographical directions in which the virus had moved. In effect, the Victorian state government reinvented the ancient system of quarantine, the benefits of which have been recorded for hundreds of years. Contact tracing began on the 25 January 2020, the same day that the first Australian case of COVID-19 was confirmed in a patient from Wuhan who had flown to Melbourne via Guangdong six days earlier. The system involved tracing all individuals who had been in close contact with the COVID-19 positive person, testing them and asking them to isolate. Details about outbreaks of infections in public spaces such as shops, cafes and bars were publicised daily on Victorian news bulletins, the DHHS website and social media. Information about the name of the site, its location, and the date and time that a COVID-19-positive customer had been present was disseminated widely and quickly. As a result, Melbournians had the choice of avoiding specific locations and having themselves tested if they had visited risky areas. Owing to the Victorian government’s PCR testing and contact tracing regimes, we knew that infection started spreading in Melbourne in mid-March and again in mid-June. Authorities were able to pinpoint specific areas and suburbs in which COVID-19 was concentrated and where the infectious clusters were moving. Public health interventions brought down not only the cases of infection but also the hospital and ICU admissions and deaths, as Figure 5 demonstrates. In 1919, by contrast, these options were not available for containing the Spanish Flu. Sydney residents learned about the spread of infection in limited areas, mainly by word of mouth, because the authorities lacked the methods of modern contact tracing or mass communication to generate a wider picture of the emerging trends in the pandemic. Attempts to keep track of daily infections in 1919 were abandoned in the early stages of the pandemic [16]. Various newspapers carried notices and information, but these were less accessible and timely than contemporary social media. In Melbourne, by contrast, health officials, politicians and the Premier himself used their twitter and other social media accounts to inform the public about the latest trends in the pandemic.

Genomic sequencing had the capacity to enhance the contact tracing system by providing authorities with information about specific strains of the COVID-19 virus and matching that information with data about the clusters of infection. Minute changes in the RNA of the coronavirus were identified by laboratory analysis and then linked to epidemiological information collected from known sites of infection. This tool sought to gather a large volume of systematic information to break the networks of viral transmission [50]. Unfortunately, the analysis of genomic clusters came too late in 2020 to prevent the hotel quarantine breaches that were identified as the origin of the pandemic’s second wave [44]. In early July, the Victorian Minister for Health announced that they had received ‘a genomic sequencing report that seemed to suggest that there seems to be a single source of infection for many of the areas that have gone across the northern and western suburbs of Melbourne’ [51]. At this stage, however, the power of genomic sequencing was not yet appreciated and contact tracing was overwhelmed by the scale of demand for tests by the public and medical professionals. These weaknesses may have contributed to the increased numbers of weekly deaths recorded in Phases 3, 4 and 5 of the pandemic (Table 2) and the dramatic mortality of the over 80-year-olds (Figure 4). Making the system more efficient probably needed each quarantine facility to house specialised units to prevent and control infection and undertake hotel-specific contact tracing [44].

Despite being slow and cumbersome, contact tracing provided essential information at a time when genomic sequencing was not an established practice and Australia lacked a surveillance organisation or system comparable to that which existed in the UK and the USA [52]. Authorities were able to justify the many mini-sequestrations and lockdowns in a way that ensured a high degree of public compliance with the restrictions. Public health officials also used information in the system to keep ahead of the pandemic by organizing two COVID-19 testing blitzes in April and June (Figure 5). The blitz announced on 25 June, for example, aimed to test 100,000 residents in 10 hotspot suburbs. Health workers went from door to door for 10 days to avoid inconveniencing residents by asking them to attend testing centres [53]. In this instance, about 10,000 people refused to be tested, largely because it was not mandatory. In any case, five days after the blitz started, Stage 3 restrictions were re-introduced, thereby reducing the mobility of untested and asymptomatic people.

The lower lethality of COVID-19 in Melbourne in 2020 relative to the Spanish Flu in Sydney in 1919 was achieved largely because of the persistence of the contact tracing system, the detailed information that it generated, the expanded testing regime that supported it and the high public awareness of the risks of infection that it created. In the worst-case scenarios, contact tracing allowed the Victorian government, fearful of the possible collapse of Melbourne’s hospital system, to justify requisitioning the use of public and private hospitals for providing the residents of aged care housing with priority access to treatment [54,55].

### 4.3. Face Masks as NPIs

The value of wearing face masks as a guard against infection was highly controversial in 1919 and again in 2020. Despite the lax use of masks, even when authorities mandated them, they played an important role in containing infection in both pandemics. Had they been used more systematically and continuously, the outcomes for the communities would have been even better, given the research that increasingly points to the considerable benefits of even cloth masks [56,57,58,59,60]. Face masks have been shown to reduce the distance travelled by the jet stream of air from a sick person; the fibrous structure of cloth masks has been demonstrated to put up multiple walls and tunnels that a virus must successfully navigate to penetrate the material, and in general, masks have been shown to reduce the viral load absorbed by a person breathing in infected air. When social distancing is not possible, masks are even more important as a barrier between the sick and non-sick. Research in China suggests that masks are important not only in crowded indoor settings but also in crowded outdoor markets where the SARS-CoV-2 can survive in the air for more than an hour and a half [61]. Cloth masks have always been and remain a useful NPI for citizens trapped in pandemic conditions, but they are imperfect because of the difficulty of firmly sealing the edge of the mask against the human face. By the best standards of air filtration in developed countries today, they have been superseded by the use of N95 masks which provide the most reliable protection [62]—provided that they are fitted properly. Surgical masks are also useful but significantly inferior to N95s [63]. However, in extreme situations where the risk of an exponential growth in infections is high, mask wearing is less effective than mini-sequestrations and lockdowns, because they allow a wide degree of discretion by the individual user.

During the Spanish Flu, the strict quarantine measures introduced into all Australian ports in October 1918 kept out the most virulent strains of the influenza virus for some four months. When influenza gripped the Australian mainland in late January 1919, swift controls were introduced by central and state governments, including the mandatory isolation of the sick and their close contacts. The New South Wales (NSW) state government introduced the ‘compulsory wearing of masks on trains, trams and ferries and in public streets, places and buildings’ [16]. However, policy flip-flops in 1919 acted as constraints on attempts to change how individuals behaved. For example, at the end of February 1919, all laws about masking were removed only to be reintroduced in late-March/early April and then again removed in mid-May, never to be introduced again despite the massive up-swing in infections [16]. By contrast, strategic interventions in Melbourne in 2020 were driven by the rise and fall in infections (Figure 5); as infection cases rose, wearing masks in public became mandatory on 22 July 2020 [64], and when they fell, mandatory mask wearing was limited to indoor areas, public transport and crowded outdoor areas. The five-month period of mandatory mask wearing from 22 July to 22 November 2020 contrasted with the much shorter duration of inconsistent mandates in Sydney in 1919 [16]. Without vaccines or antivirals, the evidence about NPIs suggests that in 1919 and 2020, the wearing of face masks was highly likely to have helped contain the spread and severity of infection in the general community. Combined with contact tracing, the wearing of face masks in Melbourne probably had a greater impact than was true of Sydney in 1919—partly because Melbournians could access a wider range and higher quality of masks in 2020.

As in China, deploying multiple NPIs in Melbourne had the additional benefit of reducing the incidence of seasonal flu [65]. ‘Year to Date’ comparisons between 2020 (on 8 August) and the previous three years show that Victoria only had 4722 flu cases relative to an annual average of 20,286 for 2017–2019—a decline of about 77% [66]. NPIs not only flattened the curve of infections for SARS-CoV-2 but also for influenza. This was evident when the NPIs were rolled back and influenza cases in Victoria began to rise dramatically. Australia’s Department of Health and Aged Care reported 35,136 cases between 1 January and 9 October 2022 [67]. The data are unequivocal: ‘Australia hasn’t made the most of the lessons of COVID-19 about masks, ventilation, and optimal vaccination to inform public health strategies to reduce the impact of seasonal flu’ [68].

The same assessment applies to the rise in COVID-19 deaths after the first year of the pandemic. In 2022, COVID-19 emerged as Australia’s ‘third leading cause of death… behind ischaemic heart disease and dementia’ [69]. In that year, COVID-19 deaths soared to 10,095 [70], not far short of the 11,552 deaths from the Spanish Flu in 1919 (Table 1). Beneath these mortality figures lurks the ever-present danger that long COVID will ultimately claim even more victims who will suffer premature deaths after years of debilitating illness. More immediately, during the first seven months of 2022, ‘non-COVID registered deaths from heart disease, cerebrovascular disease, diabetes, and dementia were between 8% and 16% higher, deaths from unspecified causes were 12% higher, and non-COVID-19 coroner-referred deaths were 7% higher than the 2015-19 baseline’ [71]. These higher-than-normal death rates have been attributed to COVID-19 [69]. Professor Doherty’s predictions in 2020, it seems, became increasingly relevant after 2020 as the pandemic entered its second, third and fourth year.

### 4.4. Study Limitations

Comparative historical studies always suffer from data problems, and in our study, the Spanish Flu epidemic generated many of these. Where possible, we have used primary sources, but secondary sources have also been cited, especially when they were published by officials in charge of the administration of vital statistics. Nevertheless, we have, of necessity, used comparative data from different sources and time periods, so there is a chance that data discrepancies may exist. The limited availability of data about the Spanish Flu in Australia cannot be fixed—we are stuck with the sources that we have used in this analysis. For example, without reliable information about the cases of infection in the community, we cannot estimate the CFR for the Spanish Flu in 1919. There are reasons to believe that the total number of cases in Sydney (about 290,000) may have been overestimated. With this number, a calculation would yield a CFR of 1.35% for the Spanish Flu. Other studies have, however, reported a CFR of about 2% [72]. The CFR for COVID-19 in Melbourne in 2020 was about 4%. The difference between the Sydney and Melbourne CFRs seems to be too large, suggesting that Armstrong [18] overestimated the extent of infection in the Sydney community. Another possibility is that the deaths recorded for Sydney were seriously understated because the medical professionals and bureaucrats could not keep up with the pace of the unfolding pandemic. However, this was probably not the case because the death and hospital admission data for Sydney were collected daily and reported as weekly mortalities, and as a result, they were more accurate than estimates about the ‘attack rate’ in the community. No such regular data collection occurred to keep track of infections. The lack of reliable information about Spanish Flu cases in Sydney is not unusual. Data about morbidity during the Spanish Flu are hard to find, as noted by Frost [72].

Another limitation of this study is how we defined and measured ‘lethality’—our approach may be disputed by others. We welcome debate about this as a way of deepening our understanding of what drives mortality and morbidity in modern times. The underlying problem in our historical comparison is that scientific knowledge about the influenza virus was lacking at the time of the Spanish Flu pandemic. In 1919, the Spanish Flu was thought to be a pneumonia-producing bacterium, with pneumococcus or streptococcus being the possible causative agents. Modern research suggests that pneumonial bacterial infection may well have contributed significantly to Spanish Flu mortality [2]. The inability of contemporaries to distinguish confidently between the Spanish Flu and other pathogens circulating in Sydney in 1919 also means that there is uncertainty about when the first cases of pandemic flu were ‘seeded’ amongst the Australian population. By contrast, we can be certain about the SAR-CoV-2 virus—the Australian Minister for Health announced on 25 January 2020 that the first case of COVID-19 had been confirmed on the basis of laboratory tests [73].

We also considered comparing the reproductive rate (R_0_) or infectivity of the Spanish Flu with COVID-19, given that this was a widely cited measure during the 2020 pandemic. However, due to the different conditions surrounding both pandemics, the comparison may not give us a valid conclusion for various reasons. Firstly, R_0_ is not a biological property of the virus but is context-specific. This is evident in the case of COVID-19, where the R_0_ value of the more recent variants would be much higher than that estimated from early data [74,75]. Secondly, the R_0_ of the Spanish Flu in 1919 is uncertain and depends on assumptions about prior immunity and other factors. In the case of COVID-19 in Australia, it is reasonable to assume that SARS-Co-2 was a previously unknown pathogen and that the population had no previous exposure or immunity. This was not true of the Spanish Flu [4,76,77]. Finally, the speed at which an outbreak takes off is indicated not only by the R_0_ but also by its incubation period. The Spanish Flu virus appears to have had a shorter incubation period than SARS-CoV-2 [25,28,29]. It is worth stressing that the R_0_ is a complex issue as it varies not only over time within a particular setting but also between different locations within that setting [78]. Such details, however, are beyond the scope of our study.

Questions might also be raised about the extent to which we can generalise from comparative historical research of the kind we have undertaken. Despite all of these limitations, we would suggest that because so much complexity surrounds attempts to understand any pandemic, historical comparisons can be invaluable in warning us about uncertainties and how to prepare for them. Who can say what problems the next pandemic might bring? What if the age-related mortality pattern of the Spanish Flu in 1919 (Figure 4) was to become a possibility? How can we plan to avoid the devastating risks that such a scenario might entail?

Some readers might ask what is the point of comparing the relatively short-lived Spanish Flu with the first year of the COVID-19 pandemic when the latter persisted for numerous years? Firstly, in 2020, Australians found themselves without any possibility of pharmaceutical responses to COVID-19. We were, in this regard, in the same situation as Australians in 1919. Secondly, we acknowledge that Professor Doherty’s warning in April 2020 that ‘COVID-19 is just as lethal as the Spanish flu’ became increasingly pertinent over the course of the COVID-19 pandemic. In 2020, lacking all other options, we chose to use NPIs, as did the citizens and authorities in Sydney in 1919. With the arrival of vaccines, antivirals and rapid antigen tests, we abandoned NPIs in 2022 and the lethality of COVID-19 surged. Perhaps the most important NPI used in 2020 was the system of contact tracing, which depended on the health authorities encouraging Victorians to get tested if they had symptoms. Reflecting on the trends that had emerged by the end of 2022, Professor Crabb declared that Australia had created the ‘worst public health disaster in living memory’ [79]. Crabb (CEO and Director of the Burnet Institute, Melbourne) [80] had predicted this shocking development in late September 2022 when he argued against the Australian National Cabinet decision (30 September 2022) to move away from ‘COVID exceptionalism’ and to treat it like any other disease by ending ‘mandatory isolation requirements for COVID-19 effective 14 October’ [81,82]. Cabinet’s decision cut across a parliamentary inquiry into the extent and impact of long COVID on Australians. There was nothing in that report to suggest that COVID-19 was like any other disease. With an estimated 370,000 Australians with severe long COVID on 31 October 2022, the inquiry recommended more government intervention, including mandatory masking and the isolation of the sick [71].

## 5. Conclusions

This study compared metropolitan Sydney and metropolitan Melbourne, the two Australian epicentres for the Spanish Flu in 1919 and COVID-19 in 2020. Our findings indicate that using various measures, the Spanish Flu appeared to be much more lethal than COVID-19. Lethality was measured by the number and rate of hospital admissions, crude death rates, age-specific death rates and ASMRs. Using these, we demonstrated the strikingly different waves of infection, their severity at various points in time and the cumulative impact of the viruses by the end of our study period, i.e., 30 September in 1919 and 2020. Hospital admissions and deaths from the Spanish Flu in 1919 were more than 30 times higher than those for COVID-19 in 2020. The ASMR per 100,000 population for the Spanish Flu was 383 compared to 7 for COVID-19—this indicates that the Spanish Flu was about 55 times more lethal than COVID-19. Our comparison of 1919 with 2020 was based on the lack of pharmaceutical interventions—this compelled communities and governments to depend on NPIs.

The lessons of 2020 and the inestimable value of NPIs remain highly relevant today, especially if we take pandemic planning seriously. We need to imagine what might happen if new, unknown pathogens or mutations of SARS-CoV-2 develop with a special appetite for killing mainly people of working age, as happened during the Spanish Flu. This, economically productive demographic cohort in contemporary Australia is far larger than the aged cohorts that have been the main victims of COVID-19 thus far. The only way to prevent elevated mortality at the start of any pandemic is to use NPIs, our first and only line of defence in 2020. Based on our experience during the first year of the pandemic in Australia, NPIs should remain our first line of defence because even after the miracle drugs and vaccines have been developed, the roll-out of pharmaceutical options is typically delayed or interrupted by the many management issues that confront health and supply systems under stress. By contrast, wearing a mask and avoiding crowded indoor and outdoor spaces requires much less individual effort: all one has to do is always carry a mask when leaving home and avoid crowds or wear a properly sealed mask if attending a mass event. The relatively small amount of behavioural change required by NPIs such as this is a worthy investment by all citizens in their own future. Such changes in the attitudes of citizens also helps to keep the option of economically and psychologically costly lockdowns at bay. Our experience with contact tracing in 2020, now powered up by genomic testing, also augurs well for the future.

Yet policy makers have much to learn from 2020, the first year of the COVID-19 pandemic in Australia. In 2020, the spread of infection was reasonably managed even with the lack of vaccines or antivirals. The abandonment of NPIs in 2022, together with the unrelenting propensity of the COVID-19 virus to mutate and assume highly infectious forms, makes the University of Michigan Medical School/US Defence Department’s warnings of 2006 relevant today:

‘In the absence of adequate stocks of an effective vaccine and/or antiviral drugs, the United States may have to rely on nonpharmaceutical interventions (NPI) to contain the spread of an infectious disease outbreak until pharmacological means become available. Because many of these NPI are costly and socially disruptive, their effectiveness and practicality need to be understood before their implementation or incorporation into a response plan’ [39].

## Figures and Tables

**Figure 1 ijerph-21-00261-f001:**
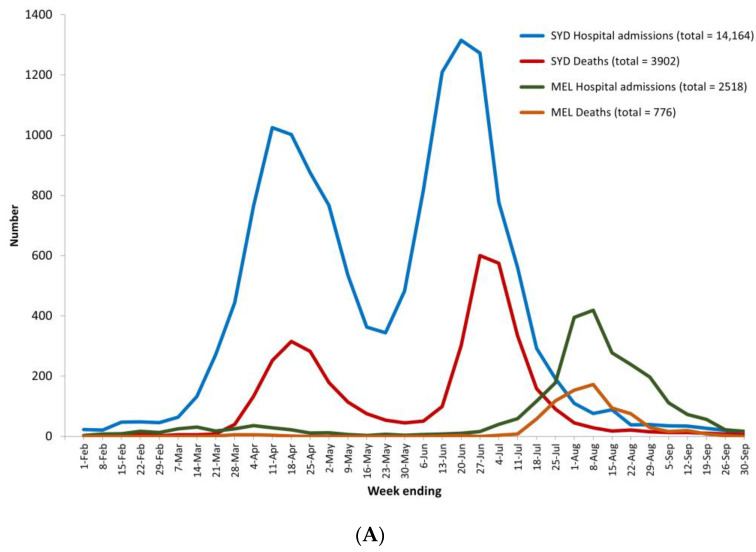
(**A**) Weekly hospital admissions and deaths in Sydney (1919) and Melbourne (2020) for weeks ending 1 February to 30 September. (**B**) Weekly hospital admissions and deaths per 100,000 population in Sydney (1919) and Melbourne (2020) for weeks ending 1 February to 30 September.

**Figure 2 ijerph-21-00261-f002:**
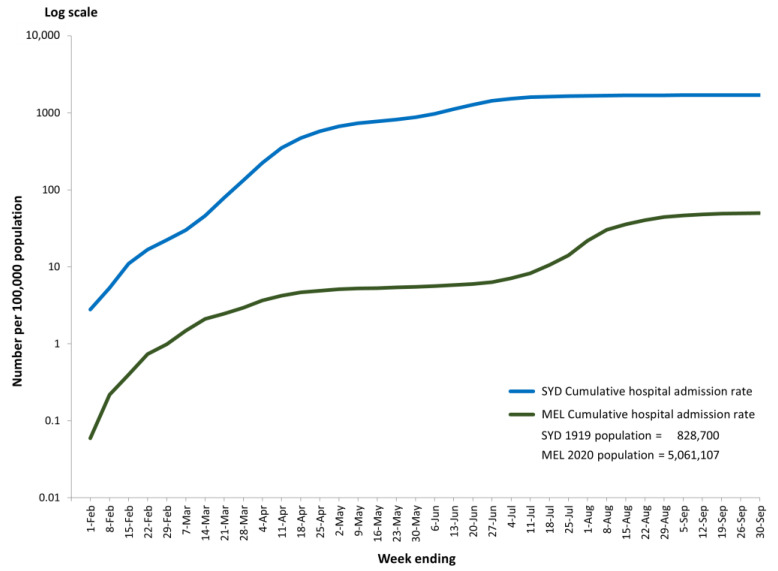
Cumulative weekly hospital admissions per 100,000 population (log scale) in Sydney (1919) and Melbourne (2020) for weeks ending 1 February to 30 September.

**Figure 3 ijerph-21-00261-f003:**
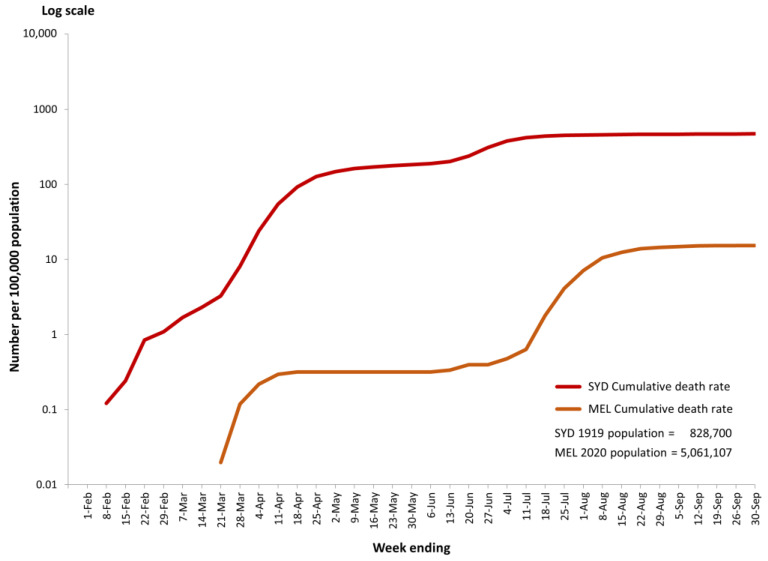
Cumulative weekly deaths per 100,000 population (log scale) in Sydney (1919) and Melbourne (2020) for weeks ending 1 February to 30 September.

**Figure 4 ijerph-21-00261-f004:**
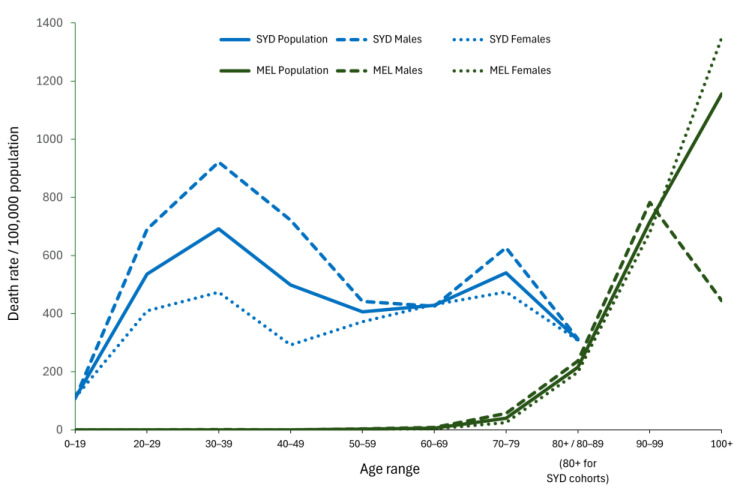
Age-specific death rates per 100,000 population from Spanish Flu in Sydney (1919) and COVID-19 in Melbourne (2020) for the period from 25 January to 30 September.

**Figure 5 ijerph-21-00261-f005:**
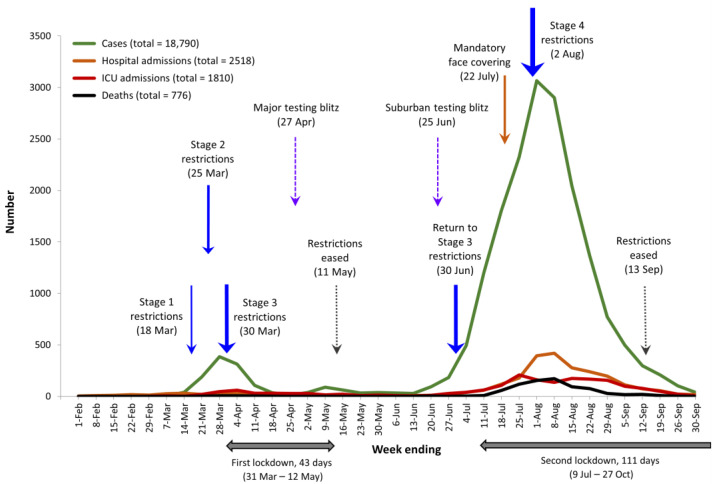
Weekly number of COVID-19 cases, hospital and ICU admissions and deaths in Melbourne for weeks ending 1 February to 30 September 2020 compared with the introduction of key NPIs.

**Table 1 ijerph-21-00261-t001:** Identifying the epicentres in Australia for the Spanish Flu in 1919 and COVID-19 in 2020.

	Spanish Flu in 1919			COVID-19 in 2020		
	Australia	Sydney	Melbourne	Australia	Sydney	Melbourne
Population [10,14]	5,217,019	828,700	743,000	25,649,248	5,295,529	5,061,107
% Capital city population in national population	100	15.9	14.2	100	20.6	19.7
% Capital city population in total state population		41.4	49.7		65.3	76.5
Number of deaths [10,11,13]	11,552 ^a^	3902 ^b^	2391 ^a^	909 ^a^	54 ^ac^	820 ^ac^
% of national deaths	100	33.8	20.7	100	5.9	90.2
Crude death rate per 100,000 population	221	471	322	3.5	1.0	16.2

^a^ Total number of deaths for the whole calendar year. ^b^ Number of deaths for the period from 1 January to 30 September 1919. ^c^ Number of deaths in New South Wales and Victoria as proxies for Sydney and Melbourne, respectively.

**Table 2 ijerph-21-00261-t002:** Phases of the 1919 and 2020 pandemics based on the periods with weekly deaths of zero, single- (1–9), double- (10–99) and triple-digit (100+) numbers.

Spanish Flu in 1919					COVID-19 in 2020				
Phases	Weekly Death Ranges	Number of Weeks	Week Ending Dates	% of 35.5 Weeks	Phases	Weekly Death Ranges	Number of Weeks	Week Ending Dates	% of 35.5 Weeks
Phase 1	0–9	8	1 February–21 March	23%	Phase 1	0	7	1 February–14 March	20%
Phase 2	10–99	1	28 March	3%	Phase 2	0–9	17 [including 8 zero weeks]	21 March–11 July	48% [22%]
Phase 3	100+	6	4 April–9 May	17%	Phase 3	10–99	1	18 July	3%
Phase 4	10–99	5	16 May–13 June	14%	Phase 4	100+	3	25 July–8 August	8%
Phase 5	100+	5	20 June–18 July	14%	Phase 5	10–99	5	15 August–12 September	14%
Phase 6	10–99	9	25 July–19 September	25%	Phase 6	1–9	2.5	19–30 September	7%
Phase 7	1–9	1.5	26–30 September	4%					
		TOTAL 35.5 weeks		100%			TOTAL 35.5 weeks		100%

## Data Availability

The raw data supporting the conclusions of this article will be made available by the authors on request.

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
