# Peer review of "Is COVID-19 as Lethal as the Spanish Flu? The Australian Experience in 1919 and 2020 and the Role of Nonpharmaceutical Interventions (NPIs)"

_ijerph, 2024, doi:10.3390/ijerph21030261_

Round 1
Reviewer 1 Report
Comments and Suggestions for Authors
Dear Editor,
The article aims to compare the effects of pandemics in 1919 and 2020 on mortality rates in two cities in Australia. The subject is of great importance and addresses a matter of public health interest for preparedness in future pandemic events. Even considering the significant challenges in retrieving historical records from a century ago, the authors present a set of cases and deaths caused by the Spanish flu. However, the manuscript lacks a more detailed analysis, incorporating statistical and epidemiological analyses that could provide greater robustness to the conclusions regarding the lethality of each pandemic.
-The manuscript employs unconventional elements for articles in health science journals, as evidenced by examples such as:
Biblical citations,
Phrases like "see discussion,"
"At the same time, we were curious to discover how...",
"Fauci expressed his concerns about the next pandemic saying that ‘we may not necessarily prevent the emergence…”,
“The trends behind these annual figures are discussed below,”
“The sheer numbers who were dying in Sydney struck terror in the hearts of all citizens.”
The structure of the text often resembles more of a book chapter than an article suitable for this journal. The manuscript delves into historical details and personal perceptions that may not necessarily be directly linked to the study's central objective. Moreover, it employs lengthy passages (and a very long Discussion) that could be condensed into shorter, more concise sentences .
-"Another data problem we faced was the lack..." – this could be addressed in the Discussion.
-The study included Deaths + prevalence of Spanish flu (36.6%) + hospital admissions of Flu + death rates (mortality rates) - the study is based on two measures of lethality – “(1) the number and rate of hospitalizations and deaths, we then diagnosed the (2) waves of infection and compared their (3) severity at various points in time and the cumulative impact of the viruses.” – (1), a Fatality Rate (number of deaths / number of cases of flu or Covid-19) could be a more accurate indicator of the severity of communicable diseases and more sensitive to the effects of non-pharmaceutical interventions in the population. It would be more suitable for comparing death rates and their impact after the implementation of non-pharmaceutical interventions.
-“Another way to express this is to say that mortality from Spanish Flu in Sydney in 1919 was almost 30 times higher or more lethal than COVID-19 in Melbourne in 2020. These results do not support the notion that the COVID-19 pandemic in 2020 was as lethal as the Spanish Flu in 1919. Professor Doherty’s statement does not appear to hold.” Comparing mortality rates from different scenarios can lead to biased conclusions – a Standardized Mortality Rate considering the age structure of the population should be used, even though it is challenging to gather such data for 1919. In this case, the Case Fatality Rate (number of deaths / number of cases) could be a more accurate indicator of the severity of communicable diseases and more sensitive to the effects of control measures in the population.
-Spatial analysis could be a valuable tool, along with its associated statistics.
-The results presented here are based on a simple comparison of rates. The study could utilize some statistical analyses to underpin the comparison, providing more robust results. I mention a few: R0 is widely used to estimate the transmission rate in epidemic events and was heavily used by the media during the Covid-19 pandemic. Several studies have estimated R0 for the Spanish flu using different methods. These values could be recalculated for Australia and compared with those estimated for Covid-19 over time, shown in a time series.
-Temporal statistics for hospital admissions could be another option (Figs. 1 and 2). One way to estimate R0 is precisely by examining the slope of the curve (exponential growth) in the time series.
-Fig 3 - waves of infection and mortality – transmission waves can be evaluated through specific statistics that consider the curve's peak, the number of infected individuals, and the transmission speed. Mortality rate may not be the most appropriate indicator of lethality here due to issues related to affected age groups and deaths from other causes.
-In this analysis, the stages of each pandemic were defined by whether the total deaths per week were reported as single, double, or triple-digit numbers for a total of 35.5 weeks. - What was the epidemiological criterion for defining the stages of the pandemics? It did not seem clear for proper understanding by the reader.
"Other NPIs such as the wearing of masks and physical distancing were, in our view, less effective because they allowed a wider degree of discretion on the part of the individual user.” Several studies have shown the effectiveness of masks in reducing Covid-19 transmission rates in various countries during the pandemic. They are effective in blocking transmission (Sars-CoV-2 can linger in the air for over an hour) and also serve as a barrier for infected individuals to release the virus into the environment.
-The discussion is excessively lengthy. It should be more concise, emphasizing the results found here and their relevance to the literature and public health. Additionally, the study's limitations were inadequately explored.
-In the manuscript's figures, the Y-axis lacks a legend.
Reviewer 2 Report
Comments and Suggestions for Authors
The manuscript provides a comparative assessment of the "lethality" of the Spanish flu in 1919, in its Australian epicentre in Sydney, to Covid-19 in the Melbourne epicentre in 2020. The research is based on historical and current data which have been carefully researched and aligned. The paper has a second part in which a detailed review of the role of nonpharmaceutical interventions is given. In a certain sense one could say that these are "two papers in one". However, the authors did meaningfully connect the two parts, and the second part adds useful context to the first. So, I am happy with that structure, and the title does also reflect the split content aproppriately.
I have two main comments on the paper, and a few minor ones.
Main comments:
- In the first part of their paper, the authors repeatedly make rather strong statements that "Professor Doherty's statement does not appear to hold", given cumulative infection, hospitalisation, and death numbers which were much higher (magnitude >30) for the Spanish flu in 1919 than for Covid in 2020. However, I think the authors would need to entertain the thought that Professor Doherty might have been talking about the risk of dying of Covid-19 once infected, sometimes referred to as case-fatality rates in the literature. Taking the numbers as given in Sections 3.1 and 3.2, one can roughly calculate these as being 1.34% for the Spanish flu compared to 4.03% for Covid, indicating that Covid is three times more "lethal" conditional on an infection.
- In the Conclusion section, the paper makes some rather strong conclusions and recommendations. However, these seem to be based on the happenings of 2022, which does not the correspond to the period which the authors have analyzed in detail, and does not correspond to the period for which good data are available (as the authors say themselves). Notably, if the Conclusion had been based on the year 2020, for which an in-depth analysis has been carried out, based on solid data, the conclusions and recommendations may have been very different? I think this is an important point that the authors need to weigh into their conclusion, especially as results as the ones presented may be picked up by the media, and could have policy implications.
Minor comments:
Table 1. Stray "." after 25,364,300
p. 4: "Given that 75.8% of the population resided in Melbourne" seems to disagree with Table 1.
Figure 1 and Figure 3, the Melbourne population given in therein seems to disagree with Table 1.
p. 12, second sentence "Instead..." is incomplete.
p.14 Sec 4.1 first sentence: This is a strange statement comparing "death rate" with "death rates and hospitalisation rates".
The statement on page 14 on the proportion of women who died from Spanish flu seems to disagree with the one at the bottom of page 4.
p 15 middle. The authors talk about "policy flip-flops" in 1919 and then say "the same applies" to Australia 2022. This is a potentially serious statement. I would like the authors to check whether this is *really* what they intend to imply.
p. 15 There are some problems with the logical flow, initially talking about a a travel ban on Feb 1, then starting a paragraph with "Prior to that" which talks about "early February".
p. 21 2nd paragraph: defied --> defined
p. 21 4th paragraph. "identified between". Remove between
p 22 last sentence "to acknowledge to information". Something sounds wrong here.
p 23 bottom: RATS --> RATs
Comments on the Quality of English Language
The English is fine.
Round 2
Reviewer 1 Report
Comments and Suggestions for Authors
Dear Editor,
All significant issues raised during the review process have been adequately addressed by the authors. The manuscript has been extensively revised and improved, providing higher quality and clarity in understanding the proposed topic. Figures and tables have been reviewed, and important points of discussion have been properly added. I believe the manuscript is now undoubtedly more suitable for the purposes of this journal.
Minor revision - some figures have borders while others do not.